# Mathematical Modeling of Mucociliary Clearance: A Mini-Review

**DOI:** 10.3390/cells8070736

**Published:** 2019-07-18

**Authors:** Ling Xu, Yi Jiang

**Affiliations:** 1Department of Mathematics, North Carolina A & T State University, Greensboro, NC 27411, USA; 2Department of Mathematics and Statistics, Georgia State University, Atlanta, GA 30303, USA

**Keywords:** mathematical modeling, cilia, mucus, clearance

## Abstract

Mucociliary clearance is an important innate host defense of the mammalian respiratory system, as it traps foreign substances, including pollutants, pathogens, and allergens, and transports them out of the airway. The underlying mechanism of the actuation and coordination of cilia, the interplay between the cilia and mucus, and the formation of the metachronal wave have been explored extensively both experimentally and mathematically. In this mini-review, we provide a survey of the mathematical models of mucociliary clearance, from the motion of one single cilium to the emergence of the metachronal wave in a group of them, from the fundamental theoretical study to the state-of-the-art three-dimensional simulations. The mechanism of cilium actuation is discussed, together with the mathematical simplification and the implications or caveats of the results.

## 1. Introduction

Mucociliary clearance is the first line of defense of the mammalian respiratory system. The mucociliary system consists of two major parts: mucus and cilia. Mucus is a thin fluid layer that coats the lung airway; cilia are hair-like organelles protruding from the cell surface and bathed in the mucus. With breathing, the mammalian respiratory tract is constantly in contact with particles in the air that could contain potentially infectious microorganisms or toxic substances. Foreign particles that land in the lung airway through airflow will be trapped in the mucous layer. The cyclic beatings of the cilia drive a unidirectional flow of mucus, which eventually move these particles out of the airway.

The past 60 years have seen many advances in the biological and physiological aspects of the mucociliary system, ranging from the molecular and cellular level to the tissue level [1,2,3,4]. Inspired by the biological discoveries, many mathematical models have been developed to explore the underlying mechanisms of the system (e.g., [5,6]). Questions that these mathematical models have attempted to address include the following: What is the force generation mechanism of the cilium, and how does it lead to the observed beating pattern? How are the cilia beating patterns coordinated to achieve the metachronal wave? Under what conditions is the metachronal wave stable? How efficient is the mucociliary transport, and how does it depend on the cilia density, beating pattern and other parameters? The complexity of the mathematical models has increased over the years, with the help of the advance in computer technology, as well as the availability of biological data and insights, such as the molecular details of an individual cilium [7,8], the multilayer composition of the mucus [9,10], the signaling between the ciliated cells and mucus [11], and the signaling between different epithelial cells [12,13]. 

This mini-review aims to highlight a few key mathematical models of the mucociliary system which are representatives of different levels of complexity. Initially, the mathematical models considered only a single cilium, and the formulations were highly simplified. Later, more biological factors were included, together with an increasing demand for computing power to solve the more complex nonlinear equations. We will discuss the theoretical hypotheses that stem from the experimental observations and how the mathematical models assist in examining these hypotheses.

The paper is organized as follows. Section 1 provides a general description of the biology and physiology of the mucociliary system. Section 2 presents a few key highlights of mathematical modeling, from hypotheses to numerical simulations, with increasing complexity. Section 3 focuses on modeling the mechanism of force generation. Section 4 summaries the paper and discusses a few possible future directions for modeling the mucociliary system.

## 2. Cilia and Mucus

Mathematical depictions of the mucociliary clearance are inspired by the structures and physiological properties of the cilia and the airway surface fluid. The human airway epithelium is composed of several phenotypes, including goblet cells that produce mucus and ciliated cells that extend cilia into the periciliary layer and mucus above (Figure 1). The mucus is in contact with the air and traps external particles inhaled into the airway. Cilia and mucus are the major components of the clearance system. The structural and mechanical properties of the cilia and the rheology of the mucus and periciliary layer are crucial in the mathematical model designs.

### 2.1. Cilia and Flagella Structure

Cilia are organelles that are widely seen in eukaryotes such as sea urchins [15] and green alga *Chlamydomonas reinhardtii* [16], where they play a vital role in sensing, food delivery, and locomotion [17,18]. Because of the remarkable similarities between cilia and flagella in both structure and motion, similar mathematical techniques have been applied to them. Therefore, we will include flagella in our discussion as well. Cilia and flagella are also present in various organisms and tissues in the mammalian body. For example, cilia are found in the pancreas [19], kidney [20], and the brain [17]. The male sperm has a long tail, and it is a flagellum [21]. Cilia in the female oviduct assist in the transport of the ovum [22]. Cilia in the animal respiratory system stop infectious pathogenic organisms and help move them out of the lung airway [23]. Cilia can be motile or nonmotile. The nonmotile cilia, also called primary cilia [24,25], are key regulators of signaling pathways during development and in tissue homeostasis [26,27]. In this mini-review, we focus on the motile cilia unless otherwise stated.

As the scanning electron micrographs of cilia (Figure 2a) and flagella (Figure 2b) show, both the cilium and flagellum are long and thin. The cilia are generally shorter than the flagella and they appear in groups, while the flagella usually appear in small numbers [28]. The motions of cilia and flagella are slightly different. The cilium is fixed at the base and sweeps back and forth, as indicated in Figure 2c; a group of cilia beating together would form a metachronal wave. The beating of a cilium consists of two strokes, effective and recovery; the pattern is asymmetric and periodic [29,30]. Empirical observations suggest that during the effective stroke, the cilium is nearly stiff, upright, and wipes fast from one side to the other; during the recovery stroke, the cilium appears soft, bends down towards the cell surface, and moves back slowly. The method by which the stiffness of the cilium is modulated remains a question. Models considering the internal cilium structure, the doublet mechanical properties, and cilium–fluid coupling (Section 2.2 and Section 3) offer some insights into possible explanations. Conversely, the base head of a flagellum is not fixed, from which a curly wave initiates and propagates to the tip (Figure 2d). This undulating wave drives the flagellum head forward, which has been quite well modeled.

Table 1 lists the properties of cilia in human and rabbit. Detailed parameters for flagella can be found in Sleigh [33].

The abilities of the cilia to propel the surrounding fluid flow and the flagella to push the base head forward are closely related to their internal structures. The axoneme is the central strand of the cilium or flagellum, and it is composed of an array of microtubules arranged in a ring with ‘9+2′ structure–nine outer doublet microtubules and two central microtubules (Figure 3). These microtubules, together with dynein arms, inter-doublet links, and sub-fibers support the typical form of a cilium or flagellum. The axoneme is crucial to the movement of the cilia and flagella, such as the initiation of motion, regulation, and behavioral responses [4,23,31,40,41,42,43]. Impaired ciliary motion typifies a few human diseases [44,45,46]. For instance, the impaired nonmotile primary cilia would cause polycystic kidney disease [47]; the disorder of the cilia causes primary ciliary dyskinesia [48], resulting in impaired transport of the mucociliary clearance.

### 2.2. Mucus and Periciliary Layer

In the mucociliary clearance system, the cilia are bathed in a fluid that coats the lung airway epithelium (Figure 1b). This fluid consists of two layers: a periciliary layer, the height of which is a little less than a typical cilium, and a mucous layer [50]. During the effective stroke, the stiff and straight cilia penetrate the mucous layer, propelling the mucus in a certain direction; during the recovery stroke, the bending cilia immerse totally in the periciliary layer, reducing the reverse propulsion.

The mucus is a complex mixture, consisting of 95% water and 5% mucins secreted from the goblet cells located between patches of ciliated cells [51,52]. The mucins of a healthy young male consist of 73% carbohydrate proteins [53], which are long, cross-link polymer chains and are responsible for the specific rheological property of the mucus. The mucus behaves as viscoelastic material [54]; in other words, the deformation of the mucus depends not only on the external force but also on the rate of the force. The periciliary layer, on the other hand, consists of mostly water and lacks the carbohydrate chains; therefore, it is often modeled as a Newtonian fluid. The air layer above the mucus is rarely considered [3,31].

In conventional models, the mucus is assumed to be a uniform blanket on top of the periciliary layer, and the periciliary layer is passively transported at the same rate as the mucus [22,45]. More recent experiments [55] suggested a novel idea that the periciliary layer behaves like a brush that promotes the movement of the mucus. Depletion of the mucus or periciliary layer leads to dysfunctional clearance and is a cause of diseases such as cystic fibrosis [56], asthma [31,57], and chronic obstructive pulmonary disease [58]. 

## 3. Phylogeny of Mathematical Models of Mucociliary System

We describe the evolution of the mathematical modeling of the mucociliary system using a few key models. These models usually have three major compartments: the motile body (cilia), the fluid flow (mucus and periciliary layers), and the bottom surface (epithelium). The techniques used in cilia motion can be applied to the flagella in general. Models for the motile body (cilium, flagellum) fall into three key categories: the over-simplified shape (rod/cylinder), a line of singular force elements (slender body theory, stokeslet), and an internal complex spring-network (Hook’s law). As with the order of these three categories, the model becomes more complex as more details of the cilium/flagellum biological structure are included. For fluid flow, the mucus is usually considered as a Newtonian or a viscoelastic network; the periciliary layer is either not included or modeled as Newtonian. The bottom surface is usually assumed to be fixed and no-slip or to be a chain of elastic springs that is deformable.

### 3.1. G1: Analytic Mathematical Formulation for Cilium/Flagellum Morphology and Motion

The early work on modeling the cilia and flagella, which we term G1 for the first generation, focused on capturing the wavy motions and tried to elucidate the mechanism of the locomotion. The axonemal structure and dynein motors were not considered in these models. Gray [59] was the first, to our knowledge, to study the undulatory locomotion in the motile body of the polychaete worm *Nereis diversicolor*. Later, Gray and Hancock [60,61] used the mathematical sinusoidal wave to describe the motion of the sea-urchin spermatozoa. Brokaw [62] further examined the bending wave along the flagellum body obtained for the spermatozoa of a sea urchin, a tunicate, and an annelid. The author stated that although the bending wave looked similar to the sinusoidal wave, it should be better depicted as a combination of circular arcs and straight lines. The author also observed that the bending wave was persistent in the flagellum’s propagation, and there might be an ‘on-or-off’ activation generator in the body elements.

Based on these observations, Brokaw proposed two bending mechanisms ([63], Figure 4a). The first one suggested that the bending was caused by the contraction of the sides of a flagellum [64]. In this case, the body-propagating wave and the local bending wave were in phase. The second one considered that the bending was induced by the shear of the internal doublets [65]. In this case, the body propagating wave and the local shear wave were in different phases. Satir [4] and Horridge [51] supported the shear mechanism in their study of the cilium bending waves. In all these models, the influence of the fluid flow on the cilium/flagellum was not considered.

### 3.2. G2: Computer-Assisted Modeling for Mechanisms of Wave Propagation

Around the same time as the analytic mathematical models, another line of study incorporated the effects of fluid flow in modeling the motile body. This approach, called the resistance theory, is a predecessor of the slender body theory. The resistance theory assumes an over-simplified shape of the motile body; e.g., a rod or a cylinder. The force exerted on the cilium or flagellum is proportional to the fluid flow velocity in the vicinity, and the force is approximated by a local resistance coefficient. Although the resistance theory ignores hydrodynamics in the problem as no feedback from the fluid flow to the motile body is considered, it has shed light on the motion of cilia and flagella with the presence of fluid flow.

Taylor [67] was a pioneer in analyzing the swimming of microscopic organisms under the influence of the fluid flow. He modeled the organism body as a long thin sheet. The temporal motion of the sheet was prescribed, and the resistance of the sheet due to the surrounding fluid was included. The swimming sheet can achieve a net translating motion. Gray [66] partitioned the flagellum into a chain of rigid short cylinders. The force on each cylinder segment was tangent to its local velocity, such that the resistance was the friction force (Figure 4b). The flagellum undulated; thus, the tangent friction force would result in a forward motion of the flagellum head. Barton & Raynor [68] modeled the cilium as a rigid rod, which was straight during the effective stroke and shortened automatically during the recovery stroke. The resistance coefficient was used to approximate the impacts from mucous flow to the cilia.

Lighthill [69] used the fact that the diameter of the flagellum/cilium was much smaller than their lengths, considered the motile body as a thin line, and introduced the slender body theory. The assumption of the slender body allows the modeling of the swimming body as a line of singular forces, which is advantageous in mathematical analysis. A ramification of this approach is the popular stokeslet model, which we will discuss later. Blake [70,71] improved the slender body theory by introducing an ‘envelope’ such that the beating cilia were replaced by a wavy ‘patch’ with the resistance force spreading over it. This method has been shown to mimic the symplectic metachronal cilia wave patterns.

Computer-assisted modeling also enabled solutions for the optimization problems. Osterman & Vilfan [72] tried to determine the optimal beating pattern based on energetic efficiency. They proposed a criterion that contained a fast and effective stroke and a slow sweeping recovery stroke. This optimization problem was then solved using the computer for both one single cilium and for a carpet of cilia. They found that the metachronal wave was crucial to achieving the highest efficiency. Eloy & Lauga [73] followed the same line of study with the goal of determining the kinematics of the most efficient cilium. In the study, the cilium was depicted as an inextensible elastic filament attached to a wall. An analytic formulation of the optimal function in the mechanical power was proposed and numerically solved. It was found that the cilium bending rigidity was essential to achieve the optimal kinematics.

Computer-assisted modeling can solve highly nonlinear analytic equations and is used as a tool to test and constrain biological hypotheses. These analytic equations are still subject to serious simplifications. For example, the full hydrodynamics were not considered, the interaction between cilia and the fluid flow was oversimplified, and the total number of cilia in the model is small, which makes it hard to explore the metachronal waves of cilia groups.

### 3.3. G3: Hydrodynamic Coupling between Cilium and Fluids

This generation of models takes the full hydrodynamics into account and is distinct from the computer-assisted models in G2. The models include intrinsic feedback among the motile body and the fluid flow, and thus better approximate the real biological environment. In addition, the usage of supercomputers has enabled simulations of hundreds of cilia with reasonable computational time. These models vary, based on their respective focuses of the mucociliary system, in levels of detail in representing the structure of the motile body, fluid flow rheology, and numerical techniques to handle hydrodynamics.

In hydrodynamics, one key parameter that discriminates different fluid flow characteristics is the Reynolds number, Re=ρLU/μ, where ρ, L, and U are the fluid density, characteristic size, and the characteristic flow velocity, respectively, and μ is the fluid viscosity. Re measures the ratio of inertial forces to viscous forces. As a frame of reference, the typical Re value for a swimming bacterium (e.g., *E. coli* in water) is in the order of $10^{-4}$, the smallest fish is about 1, a human swimmer is $10^{4}$, and a blue whale is $4\times 10^{8}$. The value of Re in the mammalian respiratory tract is around 0.01 [34], suggesting a dominating viscous force in the mucociliary system. Mathematically, the Navier–Stokes equations [74] make up the standard model that governs the fluid flow dynamics. Neglecting the inertia effect, the Navier–Stokes equations lead to Stokes equations. The stokeslet method [75] is often used to solve the Stokes equations.

#### 3.3.1. Viscous Force Alone (Re=0)

If we neglect the small inertia effect in the mucociliary system, the Reynolds number becomes zero, and the nonlinear Navier–Stokes equations are then reduced to Stokes equations [74]. This approximation yields a great convenience in computation, since the fluid flow can be represented as a superposition of stokeslets [75]. Different geometries or other constraints in the practical applications, which are difficult to solve in Navier–Stokes equations, now become more amenable using various arrangements of the stokeslets. 

The stokeslet method stems from the slender body theory, in which a set of singular force elements is postulated along the axonemal central line. Cortez [76] introduced the method of regularized stokeslets to remove the singularity. The method of regularized stokeslets is widely applied to simulate swimming micro-organelles; e.g., the helical motion of swimming [77] (Figure 5a) and bundling [78] of bacterial flagella (Figure 5b). 

Recently, Guo et al. [79] used the stokeslet method to investigate the existence of multiple modes of synchronization of the elastic micro-filaments. In their study, each filament was represented as an array of regularized stokeslets, which were positioned in such a way that the filament was inextensible and the bottom surface satisfied the no-slip boundary condition. The strength of the stokeslets can be considered as the internal bending force and tension. Such a setup can be viewed as an extension of the geometric clutch model proposed in Lindemann [80,81]. The simulations of two filaments placed side-by-side showed the bistability of the in-phase and anti-phase synchronization, indicating that the observed transition between different synchronization modes can have a dynamic explanation. The same method is also adopted by Ling et al. [82] to explore various beating patterns of one single microfilament.

Linearized Navier–Stokes equations lead to Oseen’s equation [74], and the oseenlet is its fundamental solution of the Oseen equation in free space. Efforts have been devoted to using the oseenlet to model translating objects in a viscous flow [83,84], and this approach could be a potential alternative to the stokeslet method for solving the swimming body motions in a slightly viscous fluid flow.

#### 3.3.2. Viscous and Inertial Forces (Re>0)

At Re>0, the full Navier–Stokes equations are considered. Mitran [85] introduced a model that simulated rows of pulmonary cilia in 3D. The “9+2” internal microtubule structure of an individual cilium was modeled as a finite-element beam that was curved and able to sustain a large deflection (Figure 6a). Moreover, the cilium membrane was considered to be elastic and subject to fluid stresses and internal forces transmitted from the microtubule skeleton. A two-layer fluid flow was considered: the periciliary layer was modeled using the Navier–Stokes equations and solved using the finite volume method [86], while the mucous layer was treated as a viscoelastic fluid. The model has simulated as many as 256 cilia and provided a starting example to look into the metachronal waves of cilia groups.

Yang, Dillon, & Fauci [87] explored the relationship between the cilia’s internal force generation and the resulting synchrony of cilia beating. In their study, the cilium axonemal structure was replaced by a spring network (Figure 6b). The microtubule, dynein, and nexin links were all modeled by the cross-linked springs. The internal contraction and elongation of the springs indirectly controlled the effective and recovery strokes. The mucus was assumed to follow the Navier–Stokes equation. The couplings among the cilia, mucus, and the cell surface were realized using the immersed boundary method [89]. The simulations displayed the sweeping motion of two cilia, as well as the formation of their synchrony and metachrony due to hydrodynamic couplings.

Xu and Jiang [34] aimed to identify key factors in the cilia motion that influence the ability of fluid transport in mucociliary clearance. In their simulations, the rod-propel-fluid model (Figure 6c) treated cilia as stiff rods that followed prescribed motions. This approach made it possible to separate and examine the effects of cilia density, beating frequency, and the metachronal wavelength on fluid transport. The mucous flow was governed by the Navier–Stokes equations. The cilia–fluid interactions were also handled using the immersed boundary method. The key finding of this study was that the maximum cilium height difference between strokes has the strongest effect on the net transport of fluid.

Elgeti and Gompper [88] studied the emergence of metachronal waves and introduced a mesoscopic model of 2D cilia arrays in a 3D fluid medium (Figure 6d). They focused on the stability of the metachronal waves and the resulting transport efficiency. In their model, each cilium was a semi-flexible rod that was allowed to beat independently. The cilium was activated by a bending force, determined by the neighboring fluid velocity. One highlight of this model was the inclusion of various kinds of biological noises; e.g., thermal fluctuation and molecular motors [90,91]. In their simulations, the metachronal wave has a demonstrated robustness to the biological noises.

## 4. Models of Force Generation Mechanisms

As we try to understand the single or collective beating patterns of cilia and flagella and how they change upon external stimuli, one central question is as follows: what is the mechanism of the force generation in cilium and flagellum? Three primary force generation mechanisms have been proposed for cilium and flagella: curvature-driven, internal timing, and the geometric clutch. These mechanisms are the core part of the mathematical formulations as they provide explicit ways to model motile body actuation.

The curvature-driven mechanism assumes that the entire flagellum or cilium acts as an elastic filament, possessing resistance to bending and elongation. Machin [64,92] was one of these pioneers who used mathematical formulas to study the control mechanism of the flagellum. The author hypothesized that the central line of the flagellum was composed of a series of contractile elements that were sensitive to the length and curvature changes of the flagellum. The bending at the tip of the flagellum could initiate a propagating wave. Changes in length and curvature yielded a linear tension, which delays feedback. Brokaw [65] supported the curvature-driven bending mechanism and extended the analysis by including the mechanism of microtubules sliding in the flagellum. Gueron and coworkers [93,94] applied this mechanism to a multi-cilia simulation. The authors first obtained data from one beating cilium and then incorporated the data into a two-dimensional dynamical system. The resulted simulation was able to capture essential features of the motion such as the metachronal pattern of cilia.

The internal-timing mechanism was suggested by Brokaw [95] to approximate the effects of the fluid viscosity on the behavior of sperm flagella. This mechanism says that there is an internal threshold that controls the switch point of dynein arms in the cilium and flagellum beating. This threshold is not necessarily mechanical; it could also be chemical, e.g., ATP concentration. Hill et al. [96] examined this mechanism by measuring the response in individual human airway cilia to the transmitted force. They observed that the axoneme kept switching beat direction with the same timing regardless of whether there was an external force or not. This observation rejected the assumption of a curvature-driven mechanism, at least in the case of cilia beating, and suggested that an internal timer may exist to sustain the fixed period.

The last mechanism is the geometric clutch, proposed by Lindemann [81,97,98]. The dynein arms in the axoneme generate force based on the relative displacements among microtubules. The mechanism assumed that the ‘9+2′ microtubule arrangement acted as a ‘clutch’ to turn the dynein motors on and off. In this model, bending and sliding patterns of the flagellum or cilium are consequences of dynein motors, providing a molecular regulation of the motile body.

Implementing the geometric clutch mechanism to compute the mucociliary system is not trivial as it requires a large number of dynein motors. Yang, Dillon & Fauci [87] managed to model hundreds of dynein motors and nexin links that connect microtubules (Figure 6b) as a complicated elastic spring network. Adopting the same idea of the spring network, Han and Peskin [99] allowed the dynein motors to evolve independently, each following a dynamical law for tension generation. With this improved model, the simulated 3D cilia array was able to beat spontaneously upon the fluid motion, obey cyclical oscillations, and undergo a smooth transition from synchronized motions to multiple phases. Oriola et al. [100] proposed a model using nonlinear amplitude dynamics to explore how the dynein-driven sliding yields the bending of cilia and flagella. Chakrakarti and Saintillan [101] extended the formulation of Oriola et al. [100] by including biochemical noise and hydrodynamic interactions.

## 5. Summary and Future Directions

As the famous quote by George Box states, “All models are wrong, but some are useful.” A model that encompasses every possible aspect is impossible. All mathematical models make necessary simplifications in order to focus on a certain set of features of the reality or to support certain hypotheses about the underlying mechanisms. In this mini-review, we have presented a brief survey of classical mathematical models of the mucociliary system over the last six decades, with particular attention to the cilia–fluid coupling and force generation machinery. These mathematical models were somewhat useful in helping to test hypotheses and facilitated our understanding of the fundamental mechanisms of the mucociliary system. With the advancement of technology, it gradually becomes possible to simulate a large enough number of cilia and make this physiologically relevant.

There is still a great deal of room for the improvement of the mathematical models. Most of the models of the axoneme have assumed linear elastic mechanics for the molecular structures. Such assumptions can be tested with either carefully designed single molecular mechanical experiments or by systematically screening all possible mechanics models to determine the best one. When we write the governing equations as differential equations, we already assume that the system is continuous. All molecule dynamics in the length scale of $10^{-8}$ to $10^{-5}$ were not directly described. The biochemical signaling transduction is not considered; the nexin linkages and dynein arms within the axoneme are not explicitly modeled. It would be interesting to examine the range of applicability for this continuity assumption. Also, when we start to simulate physiologically relevant carpets of cilia, it would be worth considering the spatial variations of epithelium composition as well as mucus properties in the airway. All the models eventually rely on high-quality quantitative data for validation. It would be helpful if a standardized set of data would become available as benchmarks for the modeling community.

One might also speculate how such an improved understanding of mucociliary clearance could be useful in practice. One instance could be the more effective delivery of inhaled drugs to the lung airway; e.g., an asthma inhaler or dry powder insulin inhaler. A second possibility is to help design synthetic cilia or cilia mimetics for patients with impaired lung cilia, such as magnetically or acoustically actuated cilia [102,103,104]. Synthetic molecular motors [105,106] have become available, which transfer chemical energy to motion at the nanoscale. Therefore, it is exciting to foresee the realization of grafted cilia using those molecular motors [107]. In these examples, mathematical modeling can serve as a perfect tool to examine the parameters and mechanisms characterizing the biological system and provide guidance in the engineering design; e.g., for optimal fluid transport.

## Figures and Tables

**Figure 1 cells-08-00736-f001:**
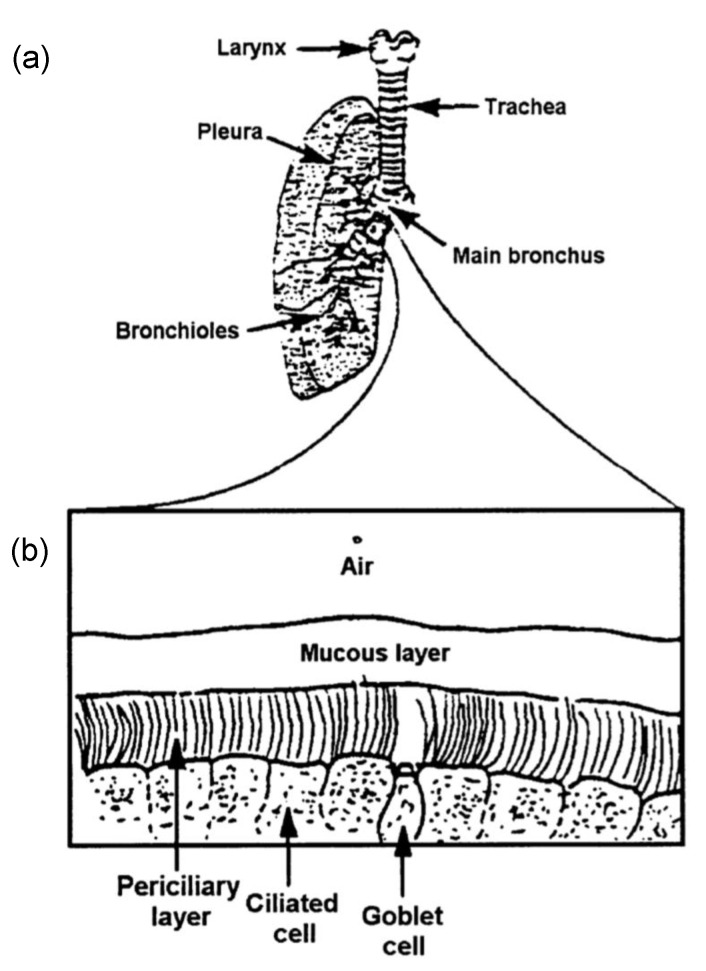
Schematic plots of the mucociliary system in the lung: (**a**) a human lung structure, (**b**) enlargement near the lung surface, from Blake [14] with permission.

**Figure 2 cells-08-00736-f002:**
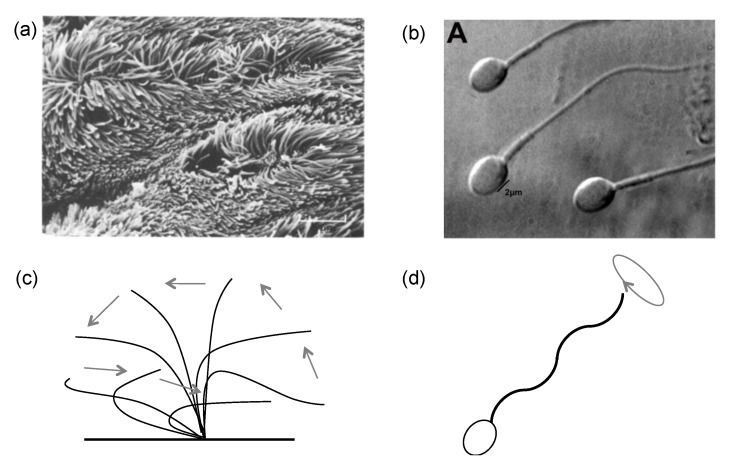
(**a**) Cilia in the rabbit tracheal, from Sanderson and Sleight [31] with permission. (**b**) Flagella of a normal spermatozoon, from Oliveira et al. [32] with permission. (**c**) The beating pattern of a cilium can be separated into an effective stroke (left arrows) and a recovery stroke (right arrows). (**d**) The beating pattern of a flagellum initiates from the base and propagates to the tip as a curly wave.

**Figure 3 cells-08-00736-f003:**
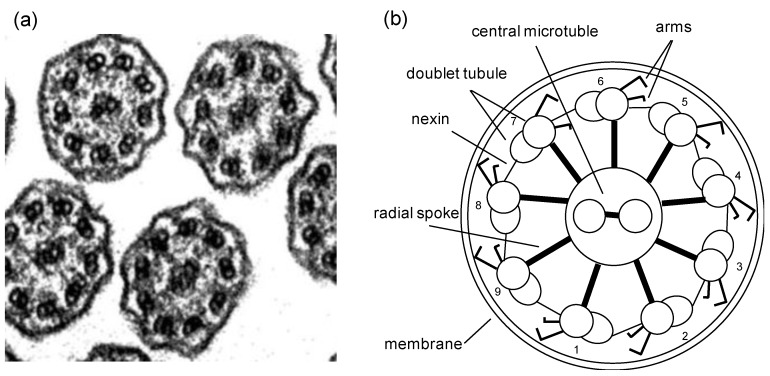
Internal structure of a motile cilium. (**a**) A transmission electron microscopic image of an axoneme, with permission from Dirksen & Satir [49]. (**b**) A schematic drawing of the “9+2” axoneme illustrating 9 microtubule doublets connected to the central two microtubules through radial spokes. Also shown are the nexin links connecting the doublets and the dynein motor arms expending from doublets.

**Figure 4 cells-08-00736-f004:**
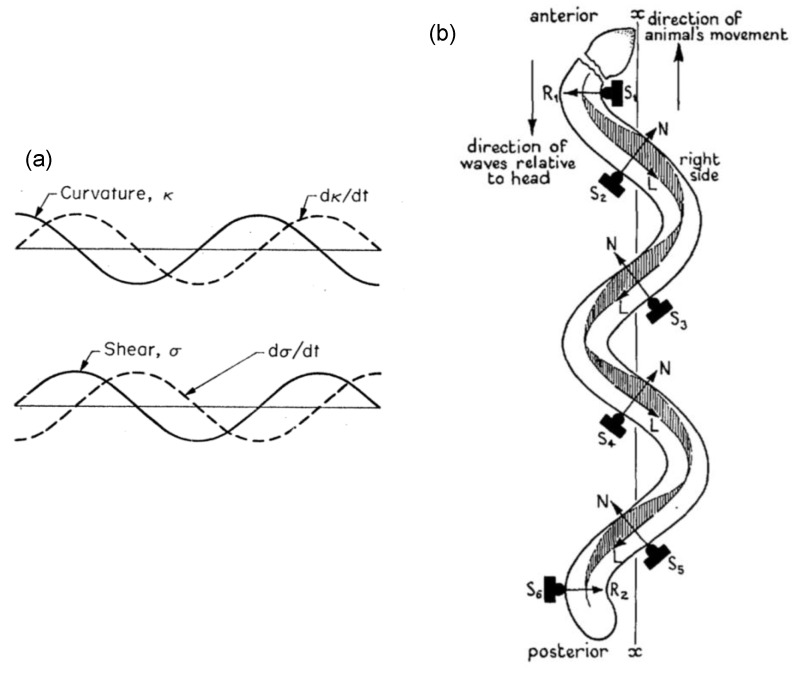
Illustration of (**a**) the curvature and shear waves approximating the flagellum undulating motion, with permission from Brokaw [63], (**b**) the tangential and normal forces (resistance forces) along the flagellum motile body, with permission from Gray [66].

**Figure 5 cells-08-00736-f005:**
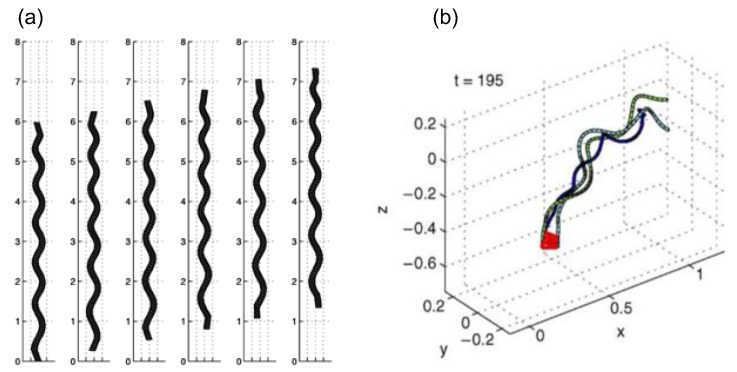
Illustration of the stokeslet method: (**a**) migration of the elastic spirochete, with permission from Cortez et al. [77], and (**b**) bundling of flagella, with permission from Flores et al. [78].

**Figure 6 cells-08-00736-f006:**
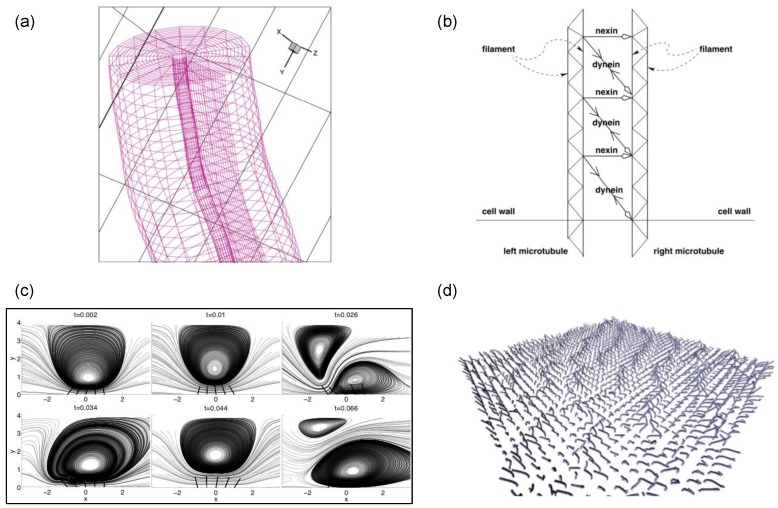
(**a**) The cilium modeled as a finite element beam and the nearby body-fitted grid, from Mitran [85] with permission; (**b**) the cilium modeled as an elastic spring network, from Yang, Dillon and Fauci [87] with permission; (**c**) streamlines in the fluid flow induced by five cilia (rods) at a sequence of times, from Xu and Jiang [34] with permission; (**d**) snapshot of an large array of beating cilia, from Elgeti and Gompper [88] with permission.

**Table 1 cells-08-00736-t001:** Parameters of cilia that are important for mathematical modeling. Courtesy of Xu and Jiang [34].

Parameter	Values	References
Cilium length	5–7 μm	Sanderson & Sleigh [31]
Cilium density	6–8 μm^−2^	Sleigh et al. [23]
Beating frequency	13–29 Hz	Sanderson & Sleigh [31]
	14 Hz	Low et al. [35]
	15.6 Hz	Marino & Aiello [36]
	11–1-5 Hz (human nasal cilia without mucus)	Chilvers & Challaghan [37]
Tracheal mucus velocity	5.5 mm/min^−1^	Foster et al. [38]
	6.7–11.4mm/min^−1^	Friedman et al. [39]

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
