# Peer review of "Mathematical Modeling of Mucociliary Clearance: A Mini-Review"

_cells, 2019, doi:10.3390/cells8070736_

Round 1

Reviewer 1 Report

The paper gives a useful review of recent research on developing mathematical models of cilia and flagella. I recommend the paper is published after the following issues are addressed:

“The periciliary layer, on the other hand, consists of mostly water and lacks the carbohydrate chains, therefore, it behaves like an elastic material exhibiting a linear response to the external force.” This sentence is contradictory. Did you mean the periciliary layer behaves as a Newtonian liquid if it mostly consists of water?

I suggest switch the order of the presentation of chapters 2.3.1. and 2.3.2. First discuss the models utilizing the Stokes equation and then those solving the full Navier Stokes. Furthermore, it is somewhat misleading to characterize models as with and without inertia. Although it is technically correct it also may confuse the reader by imply that inertia is important, whereas it is not the case.

Including a table with mechanical properties of cilia and mucus will be useful.

“All molecule dynamics in the length scale of 10^-8 to 10^-5 are ignored.” This statement is misleading since the molecular dynamics gives rise to the bulk material properties and transport coefficients used in the continuous approximations.

Another area of future research can be related to organ-on-a-chip applications of cilia and cilia mimetics. In this context, the development of models that are suitable for modeling of cilia mimetics, such as recently demonstrated magnetically actuated cilia, would be of interest and useful to briefly discuss in the review and provide relevant references.

Author Response

We thank the reviewer for the insightful comments and critiques on the paper. We have addressed all comments in the revised paper, as detailed below. The reviewers’ comments are listed in black, followed by our response in blue, with the changes in italic blue.

Review #1

1.     “The periciliary layer, on the other hand, consists of mostly water and lacks the carbohydrate chains, therefore, it behaves like an elastic material exhibiting a linear response to the external force.” This sentence is contradictory. Did you mean the periciliary layer behaves as a Newtonian liquid if it mostly consists of water?

The periciliary layer is often modeled as a Newtonian fluid, which of course is a simplification by ignoring all non-water ingredients. The Newtonian fluid flow is isotropic and is described by a linear constitutive equation between the velocity gradient and shear force (Fung, 1969, eqn 7.3-3). That's why we described the periciliary layer as a linear material. We understand the confusion and have modified the text (page 6, sec 1.2, para. 2) to "The periciliary layer, on the other hand, consists of mostly water and lacks the carbohydrate chains; therefore, it is often modeled as a Newtonian fluid."

Fung Y.C. (1969) First course in continuum mechanics, Englewood Cliffs, N. J. Prentice-Hall

2.     I suggest switch the order of the presentation of chapters 2.3.1. and 2.3.2. First discuss the models utilizing the Stokes equation and then those solving the full Navier Stokes.

Thanks for the suggestion – we switched the order of chapters 2.3.1 and 2.3.2, to discuss the models using the Stokes equation before those solving the full Navier-Stokes equations.

3.     Furthermore, it is somewhat misleading to characterize models as with and without inertia. Although it is technically correct it also may confuse the reader by imply that inertia is important, whereas it is not the case. Including a table with mechanical properties of cilia and mucus will be useful.

We have changed “Without inertia” to “Viscous force alone (Re=0)”, “With inertia” to “Viscous and inertial forces (Re>0)” to avoid confusion.

4.     “All molecule dynamics in the length scale of 10^-8 to 10^-5 are ignored.” This statement is misleading since the molecular dynamics gives rise to the bulk material properties and transport coefficients used in the continuous approximations.

We agree with the reviewer that molecular dynamics gives rise to the bulk material property. To avoid confusion, we have changed the sentence to “All molecule dynamics in the length scale of 10^-8 to 10^-5 are not directly described.”

5.     Another area of future research can be related to organ-on-a-chip applications of cilia and cilia mimetics. In this context, the development of models that are suitable for modeling of cilia mimetics, such as recently demonstrated magnetically actuated cilia, would be of interest and useful to briefly discuss in the review and provide relevant references.

We thank the referee for this interesting suggestion.  We added the examples of magnetically actuated cilia (Gauger et al. 2009, Meng et al. 2019) and acoustically actuated cilia (Orbay et al. 2018) in “Summary and future directions”. In both cases numerical studies of artificial cilia can help optimize fluid transport.  

Reviewer 2 Report

In the minireview 'Mathematical modeling of mucociliary clearance', the authors provide a summary of literature relevant to, on one hand, the focused problem of mucociliary dynamics in the airway and more broadly, the question of ciliary/flagellar propulsion.  The provided manuscript seems to concentrate on (roughly) pre-2010 literature and has few references less than 5 years old; this is somewhat misleading as mucociliary dynamics remains an active research area (there was a 2019 Gordon conference on this problem, for example). Even so, the authors do provide a fairly comprehensive review of 'older' literature.

One persistent issue I have with this manuscript is the repeated conflation of motile cilia and sensory cilia. The authors use the term 'cilia' to exclusively refer to motile cilia, so statements like 'Cilia are motile organelles that are widely seen in...' (first sentence, section 1.1) are technically false.  Similarly, "Impaired ciliary motion" (page 4) may typify a number of diseases, but Polycystic Kidney Disease is not one of them, as the kidney does not have motile cilia.  Lastly, the authors do not mention nodal cilia at all- I would agree they are not relevant to their review, but again, their blanket use of the term 'cilia' obliterates any distinctions.

Another potential issue I have is that the authors restrict their discussion to the use of 'Stokelets', when 'Oseenlets' are also used to model slender body fluid-structure interactions.  Oseenlets may actually be a significant improvement, and so I wonder why the authors do not mention them.

In addition, there are other statements that are either confusing or not correct, for example: 

(page 3, near the bottom) "During the effective stroke, the cilium is nearly stiff...".  This sentence essentially states that cilia modulate their own mechanical properties during a stroke ('becoming soft'), yet provides no insight into how this could be modeled or accomplished.

(first paragraph, section 1.2) "During the effective stroke, the stiff and straight cilia penetrate into the mucus layer,...".  Again, I don't think this is a settled issue, but if it is, the authors need to provide some references.  More generally, the discussion about stratified airway fluids (periciliary layer, mucus, air) is somewhat confusing- the regulation of periciliary fluid thickness is not fully understood. And on page 4, the perciliary layer, being a Newtonian fluid, does not behave as an elastic material but rather as a viscous material.

Some minor comments about use of English language: There are numerous grammatical and spelling errors that need to be fixed.  A few examples:

Table 1: 'gracheal mucouss'

Page 4: 'dyne' arms, 'Mucocliary' clearance, 

Author Response

We thank the reviewer for the insightful comments and critiques on the paper. We have addressed all comments in the revised paper, as detailed below. The reviewers’ comments are listed in black, followed by our response in blue, with the changes in italic blue.

Review #2

1.     In the minireview 'Mathematical modeling of mucociliary clearance', the authors provide a summary of literature relevant to, on one hand, the focused problem of mucociliary dynamics in the airway and more broadly, the question of ciliary/flagellar propulsion. The provided manuscript seems to concentrate on (roughly) pre-2010 literature and has few references less than 5 years old; this is somewhat misleading as mucociliary dynamics remains an active research area (there was a 2019 Gordon conference on this problem, for example). Even so, the authors do provide a fairly comprehensive review of 'older' literature.

Thanks for the suggestions, we have added a few more references that appear in the last three years.

·      In Section 2.3.1 of “Viscous force alone, (Re=0)”, we added one more reference that uses stokeslet method to investigate the multiple beating mode of microfilaments.

Ling F., Guo H., Kanso E. (2019) Instability-driven oscillations of elastic microfilaments, J. R. Soc. Interface, 15: 20180594

·      In Section 3 of “Models of force generation mechanisms”, we added a couple nonlinear amplitude dynamics models, which were built on the Geometric Clutch model.

Oriola D., Gadelha H., and Casademunt J. (2017) Nonlinear amplitude dynamics in flagellar beating, R. Soc. Open sci., 4: 160698

Chakrabarti B., Saintillan D. (2019) Spontaneous oscillations, beating patterns, and hydrodynamics of active microfilaments, Phys. Rev. Fluids, 4: 043102

·      In Section 4 of “Summary and future direction”, we added the new examples on magnetically and acoustically actuated artificial cilia.  

Meng F., Matsunaga D., Yeomans, J.M. Golestanian R. (2019) Magnetically-actuated artificial cilium: a simple theoretical model, Soft Matter, 15: 3864–3871

Orbay S, Ozcelik A, Bachman H, Huang T.J. (2018) Acoustic actuation of in situ fabricated artificial cilia, J. Micromech. and Microeng., 28 (2): 025012

2.     One persistent issue I have with this manuscript is the repeated conflation of motile cilia and sensory cilia. The authors use the term 'cilia' to exclusively refer to motile cilia, so statements like 'Cilia are motile organelles that are widely seen in...' (first sentence, section 1.1) are technically false. Similarly, "Impaired ciliary motion" (page 4) may typify a number of diseases, but Polycystic Kidney Disease is not one of them, as the kidney does not have motile cilia. Lastly, the authors do not mention nodal cilia at all- I would agree they are not relevant to their review, but again, their blanket use of the term 'cilia' obliterates any distinctions.

                            Thanks for the reviewer to point out the ambiguity of the term ‘cilia’.

Page 3, Sec1.1, first sentence, we removed the word motile, now it becomes “Cilia are organelles that are widely seen …”

Page 3, Sec 1.1, last sentence is added, “Cilia can be motile or nonmotile. The nonmotile cilia, also called primary cilia (Sorokin 1968), are key regulators of signaling pathways during development and in tissue homeostasis (Satir 2010). In this minireview, we focus on the motile cilia unless otherwise stated.

Page 5, 3nd line above Fig.3, about the polycrystic kidney disease, the sentence is “… the impaired nonmotile primary cilia would cause the polycystic kidney disease…”, which correctly referred to primary cilia.

                         We have checked and modified the term “motile cilia” throughout to eliminate this confusion. 

3.     Another potential issue I have is that the authors restrict their discussion to the use of 'Stokelets', when 'Oseenlets' are also used to model slender body fluid-structure interactions. Oseenlets may actually be a significant improvement, and so I wonder why the authors do not mention them.

Thanks for the suggestion. We have added one paragraph at the end of section 2.3.1 about the Oseenlet, “Linearizing Navier-Stokes equations yields the Oseen’s equation (Batchelor 1967), and the Oseenlet is its fundamental solution in free space.  Efforts have been devoted to using the Oseenlet to model translating objects in a viscous flow (Price & Tan 1992; Lu & Chwang 2005), and this would be a potential alternative method to stokeslet for solving the swimming body motions in a slightly viscous fluid flow.

Price W. G., Tan M. (1992) Fundamental viscous solitons or transient oseenlets associated with a body manoeuvring in a viscous fluid, Proc. R. Soc. Lond. A., 438: 447-466

Lu D. Q., Chwang A. T. (2005) Unsteady free-surface waves due to a submerged body moving in a viscous fluid, Phys. Rev. E, 71: 066303

4.     In addition, there are other statements that are either confusing or not correct, for example: (page 3, near the bottom) "During the effective stroke, the cilium is nearly stiff...". This sentence essentially states that cilia modulate their own mechanical properties during a stroke ('becoming soft'), yet provides no insight into how this could be modeled or accomplished. (first paragraph, section 1.2) "During the effective stroke, the stiff and straight cilia penetrate into the mucus layer,...". Again, I don't think this is a settled issue, but if it is, the authors need to provide some references.

We agree with the reviewer that the cilia seem to modulate their own stiffness during a stroke, but the mechanisms underneath this modulation during the strokes remains not well understood. Although often not state explicitly, discovering such mechanisms has been one main objective of many mathematical modeling efforts for mucociliary clearance. For example, Yang, Dillon, Fauci (2008) modeled the axoneme as a spring network. During the effective and recovery strokes, the intracellular structure and mechanical property of springs are modified either passively based on the sliding between pairs of outer doublets or actively based on the dynein arm activation. In fact, the entire section 3 of “Models of force generation mechanism” is devoted to exploring how cilia/flagella regulate their motion via adjusting their structure and internal mechanical property according to the surrounding environment. 

We added the following statement in Introduction: “How the stiffness of the cilium is modulated remains a question. Models considering the internal cilium structure with mechanical properties and cilium-fluid coupling (sections 2.3 and 3) offer some insights into possible explanations.”

5.     More generally, the discussion about stratified airway fluids (periciliary layer, mucus, air) is somewhat confusing- the regulation of periciliary fluid thickness is not fully understood.

To the best of our knowledge, the mucociliary clearance system considers primarily two components: mucus and cilia (Sanderson & Sleigh 1981; Houtmeyers et al. 1999).  Thus, the mathematical model usually does not consider the air flow above. 

Height of the periciliary layer is typically a little less than the cilia (Houtmeyers et al. 1999). The effect of thickness change in the periciliary fluid have not been explored extensively in mathematical models.

          Sanderson M. J., Sleigh M. A. (1981) Ciliary activity of cultured rabbit tracheal epithelium: beat  pattern and metachrony, J Cell Sci., 47:331–347

Houtmeyers E., Gosselink R., Ramirez G.G., Decramer M. (1999) Regulation of mucociliary clearance in health and disease, Eur. Respir. J. 13: 1177–1188

6.     And on page 4, the perciliary layer, being a Newtonian fluid, does not behave as an elastic material but rather as a viscous material.

The Newtonian fluid flow assumes isotropic and a linear constitutive equation between the velocity gradient and shear force (Fung, 1969, eqn 7.3-3). That's why we describe the periciliary layer as a linear material.

We have made the following revision in section 1.2 at the end of para.2, "The periciliary layer, on the other hand, consists of mostly water and lacks the carbohydrate chains; therefore, it is often modeled as a Newtonian fluid."

Fung Y.C. (1969) First course in continuum mechanics, Englewood Cliffs, N. J. Prentice-Hall

7.     Some minor comments about use of English language: There are numerous grammatical and spelling errors that need to be fixed. A few examples: Table 1: 'gracheal mucouss', Page 4: 'dyne' arms, 'Mucocliary' clearance.

Thanks very much! We have carefully eliminated typos and grammar mistakes.

Round 2

Reviewer 2 Report

The authors have fully addressed my prior comments, thanks- this is a nice review!